# The Advanced Floating Chirality Distance Geometry Approach―How Anisotropic NMR Parameters Can Support the Determination of the Relative Configuration of Natural Products

**DOI:** 10.3390/md18060330

**Published:** 2020-06-24

**Authors:** Matthias Köck, Michael Reggelin, Stefan Immel

**Affiliations:** 1Alfred-Wegener-Institut für Polar-und Meeresforschung in der Helmholtz-Gemeinschaft, Am Handelshafen 12, 27570 Bremerhaven, Germany; 2Clemens-Schöpf-Institut für Organische Chemie und Biochemie, Technische Universität Darmstadt, Alarich-Weiss-Straße 4, 64287 Darmstadt, Germany; re@chemie.tu-darmstadt.de

**Keywords:** chirality, configurational analysis, distance geometry, NMR spectroscopy, NOE data, residual dipolar couplings

## Abstract

The configurational analysis of complex natural products by NMR spectroscopy is still a challenging task. The assignment of the relative configuration is usually carried out by analysis of interproton distances from NOESY or ROESY spectra (qualitative or quantitative) and scalar (*J*) couplings. About 15 years ago, residual dipolar couplings (RDCs) were introduced as a tool for the configurational determination of small organic molecules. In contrast to NOEs/ROEs which are local parameters (distances up to 400 pm can be detected for small organic molecules), RDCs are global parameters which allow to obtain structural information also from long-range relationships. RDCs have the disadvantage that the sample needs a setup in an alignment medium in order to obtain the required anisotropic environment. Here, we will discuss the configurational analysis of five complex natural products: axinellamine A (**1**), tetrabromostyloguanidine (**2**), 3,7-*epi*-massadine chloride (**3**), tubocurarine (**4**), and vincristine (**5**). Compounds **1**–**3** are marine natural products whereas **4** and **5** are from terrestrial sources. The chosen examples will carefully work out the limitations of NOEs/ROEs in the configurational analysis of natural products and will also provide an outlook on the information obtained from RDCs.

## 1. Introduction

The determination of the relative and absolute configuration of natural products is essential to understand their interactions in the biological field and to allow their procurement through total synthesis. The structure determination of natural products by NMR spectroscopy [1,2,3] is usually divided into two more or less “independent” approaches: (a) constitutional assignment and (b) configurational and conformational assignment (see Figure 1). The constitutional assignment will not be covered in the present manuscript. We will focus on the discussion of the assignment of the relative configuration and conformation only.

### 1.1. NOEs/ROEs in Structure Elucidation

So far, there is no general NMR method for a secure assignment of the relative configuration of non-crystallizable natural products [4,5,6]. Valuable information is provided by NOEs or ROEs which allow to derive actual interproton distances by volume integration of the cross-peaks in the NOESY or ROESY spectrum. The H, H distances are obtained by the comparison of the peak volume with a cross-peak of known distance (the so-called calibration or reference peaks). The determination of the relative configuration from NOE- or ROE-derived interproton distances can be accomplished in different ways [3]. In the past, this was mainly carried out in a qualitative way using molecular-mechanics or density functional theory (DFT) derived structure models. In particular, the DFT approach is restricted to relatively small systems because these types of calculation quickly become prohibitively expensive for larger structures or large numbers of diastereomers that need to be considered.

Another possibility would be to run rEM (restrained energy minimization) [7] or rMD (restrained molecular dynamics) [8,9] simulations for all possible relative configurations, and generally, the one with the lowest error with respect to the experimental restraints is chosen as the correct relative configuration of the investigated molecule. The disadvantage of this approach is that it is very time consuming, especially for molecules with many unknown stereogenic centers because for every diastereomer separate simulations need to be run (2*^n^*^−1^ calculations), albeit these may be automated in computer-assisted structure elucidation protocols [10,11]. However, MD simulations are biased by the choice of the force-field (for uncommon structural fragments these might even lack appropriate parameters at all) and the user’s choice of the initial geometry (“starting configuration and conformation”). Relative conformational energies obtained from DFT calculations may be inaccurate up to ~1–2 kcal/mol^−1^ (amounting to errors in Boltzmann weights of conformers differing by factors of ~0.20–0.03 at 300 K!) depending on the treatment of electron-electron correlation and/or dispersion interactions [12].

One method of choice for small molecules with several stereogenic centers is the combination of distance geometry (DG) [13,14,15,16] and distance bounds driven dynamics (DDD) calculations using NOE/ROE-derived distance restraints (r) [3,5,16,17,18,19]. The most important aspects of the NOE/ROE-restrained DG/DDD method (rDG/DDD) is the possibility to allow configurations to dynamically change during the simulation (floating chirality, fc) and therefore to determine the conformation and the relative configuration of small organic molecules simultaneously (fc-rDG/DDD). The DG approach (see Figure 2) considers holonomic distance restraints as lower ($d_{min}$) and upper ($d_{max}$) bounds of atom-atom distance relations, which are derived from the molecular constitution (which must be known!), as well as 1,2- (bonds), 1,3- (angles), and 1,4-connectivities (torsions) and experimental NOE/ROE-derived restraints can be added to this set of limits. Within these restraints, structure models are generated solely based on distance information, removing the bias to any initial input reference model, and these models are further refined in a simulated annealing approach. Chirality is incorporated in the DG approach using signed chiral volumes, which basically describe the volume enclosed by the substituents on tetrahedral centers, and which simultaneously encode for opposite configurations through sign inversion (see Section 4).

The concept of floating chirality (fc) was introduced for the assignment of diastereotopic protons or methyl groups in proteins. This approach was first applied in 1988 to distance geometry (DG) calculations [20] and in 1989 to rMD simulations [21]. In DG calculations, floating chirality is achieved by not using chiral restraints (chiral volumes) for unknown prochiral and stereogenic centers, whereas in rMD, floating chirality is achieved by reducing or removing the force constants of the angles which define the chiral centers. Even more, DG uses no energy penalty or additional out-of-plane terms to guarantee that the full set of permutations for all stereogenic centers is generated. In general, DG uses a single chiral volume restraint on one selected stereogenic center only in order to avoid enantiomeric configurations (see discussions below). However, in contrast to rMD simulations, DG does not use any physical force-field of any type, and thus removes any intrinsic bias imposed on the results by this choice. DG relies solely on experimental data like distances between atoms or anisotropic data (see below) and all stereogenic centers are allowed to adopt their relative configuration in accordance with the experimental data.

Moreover, the strength of the DG approach is that all structure models are first generated in four-dimensional (4D) space before these are transferred into “real” 3D space. The extra dimension provides additional degrees of freedom to assemble structures of different configurations and conformations within the limits of the distance bounds. Most notably, the sequence of 4D and 3D simulated annealing steps has major benefits for the robustness and quality of configurational sampling, as inversions of 3D objects (e.g., stereogenic centers) become simple rotations in 4D, and thus the “energy” barriers between alternate diastereomers are effectively lowered or even removed altogether (see Figure 2, and in the Section 4, Figure 12).

NMR-derived experimental data such as NOE/ROE distances, scalar couplings, residual dipolar couplings (RDCs), residual quadrupolar couplings (RQCs), and residual chemical shift anisotropies (RCSAs) can be incorporated in this DG approach. Here, all experimental parameters are accounted for as sums of squared violations $\Delta X^{2}={\left(X^{exp}-X^{calc}\right)}^{2}$ of experimental versus back-calculated values, and these deviations are added up in a harmonic approximation as pseudo energy terms $E=1/2K\sum \Delta X^{2}$ with empirical force constants K. In total, the sum of these terms based on NMR data, and violations of distance bounds or, if applicable chiral volume restraints, define a dimensionless total penalty or pseudo energy function, which must not be confused with a MD- or DFT-derived “real” molecular energy, and the lower this pseudo energy penalty becomes, the better the restraints based on experimental data are fulfilled. A comprehensive description of all energy terms is given in the Section 4. In this context, these violation energies, and in particular their partial derivatives $\partial E_{total}/\partial r$ with respect 4D and 3D Cartesian atomic coordinates, are considered as forces which drive the structure evolution in a simulated annealing type approach–and thus the structures evolve from the data rather than being evaluated against pre-calculated structures only.

Up till now a general application of the DG approach to all different kind of natural products was hindered by the fact that NOEs/ROEs cover only short-range interactions (up to 400 pm for small molecules) and was hampered or even impossible for proton-deficient structures. This can now be overcome by the use of anisotropic NMR parameters (RDCs, RQCs, and RCSA) in the structure under investigation.

### 1.2. RDCs in Structure Elucidation

In contrast to NOEs/ROEs, residual dipolar couplings (RDCs) are anisotropic NMR parameters, which are global in nature and independent from the distances between the vectors connecting the coupling nuclei. RDCs, RQCs, and RCSAs are NMR observables that can now be used within the fc-rDG/DDD method using the recently published software *ConArch*^+^ [22,23]. Within this investigation, only the usage of RDCs will be discussed here.

Standard NMR investigations are carried out in isotropic solutions, where usually the dipolar couplings are averaged out by isotropic tumbling of the molecules. If this is not the case, either by the presence of paramagnetic metal ions [24] or anisotropic susceptibility of diamagnetic macromolecules [25] or, more general, the presence of an anisotropic medium, the molecules will be partially oriented with respect to the external magnetic field, and residual dipolar couplings (RDCs) can be measured (detailed reviews can be found at [26,27,28,29,30,31]). An anisotropic environment is generated by an alignment medium (AM), examples for AMs are stretched gels [32,33,34,35,36,37,38,39,40] or lyotropic liquid crystalline (LLC) phases [41,42,43,44,45,46,47,48,49].

The size of ^1^*D*_CH_ RDCs depends on the time-averaged orientation of the CH-vector and its averaged angle with respect to the external magnetic field $B_{0}$ (see Figure 3). The one-bond (CH) dipolar coupling is usually obtained by comparison of HSQC-type experiments run in isotropic and anisotropic environment [50,51]. A very popular variant of these HSQC experiments is the so-called CLIP/CLAP-HSQC [52], which is run without F_2_ decoupling in order to observe the one-bond coupling in F_2_. The residual dipolar coupling adds to the scalar coupling leading to a total coupling constant (^1^*T*_CH_) from which the residual dipolar coupling (^1^*D*_CH_) can be calculated (^1^*T*_CH_ = ^1^*J*_CH_ + 2 ^1^*D*_CH_).

Analysis of RDC data is less straightforward than the interpretation of isotropic data such as chemical shifts and scalar coupling. However, given a molecular geometry for the compound analyzed, RDCs can be back-calculated from the experimental data and this structural model in a parameter-free fashion using natural constants only, and an alignment tensor can be computed which describes the average orientation of the molecule in relation to the magnetic field [53]. Frequently, alternative relative configurations of analytes imply different relative orientations of CH vectors, and thus RDCs are very sensitive configuration probes even for cases, where stereogenic centers are separated by many bonds. Usually, the configuration which displays the best correlation between experimental and back-calculated RDC data ($D^{exp}$ vs. $D^{calc}$) is considered as the correct one (see Figure 3). RDC analysis is based on the assumption that the chemical shifts in the isotropic and anisotropic phases do not change or change only slightly. The standard procedure does not include a re-assignment of the molecule under study in the anisotropic phase, but the assignment could be questionable if larger changes in the chemical shifts are observed.

However, crucial for the interpretation of RDC data is the fact that accurate structure proposals must be provided at first hand, which are then evaluated against the experimental NMR data, and a thorough error analysis has to be carried out in order to ascertain configurational assignments [23]. The necessity for pre-evaluation of conformational preferences may become problematic for flexible or larger molecules. Moreover, this type of analysis has to be repeated for all $2^{n-1}$ diastereomers if the molecule contains n stereogenic elements. In a recent report [22], we have demonstrated on how to include RDC information in DG simulations in both 4D and 3D space, using a pseudo energy penalty function $E_{RDC}=1/2K_{RDC}\sum {\left(D^{exp}-D^{calc}\right)}^{2}$ similar as described above. This now provides the advantage that the prerequisite of the beforehand structure generation is dropped altogether. Instead, the correct configuration emerges from these RDC-driven rDG types of simulations as a direct consequence and within the boundaries of these experimental restraints.

Though the mathematical details for the treatment of NOEs/ROEs and RDCs differ vastly, the pseudo energy error function allows to arbitrarily combine these different types of restraints within DG, and structures are generated fulfilling all experimental parameters best. However, there is one additional fundamental difference between NOE and RDC data. For the former, only a single NOE “data set” can be obtained, whereas for the latter RDCs multiple “data sets” can be obtained when measuring the NMR data under different alignment conditions (i.e., different alignment media [23,54,55,56,57], multi-component multi-phase AM [46], temperature dependent AM [43,58], etc.). Though this might entail considerable experimental effort, these multi-alignment data sets can also be exploited in the DG implementation of ConArch^+^ [22]. Under the assumption that the conformational preferences of the analyte do not change significantly for alternate alignment conditions, different sets of RDCs can provide crucial additional and independent structure information, which may contribute significantly to the certainty with which configurational assignments are supported by experimental data [23,54,55,56,57].

In the sequel, the application of the fc-rDG/DDD method will be demonstrated on five complex natural products (see Scheme 1). The dimeric cyclic pyrrole-imidazole alkaloid (PIA) axinellamine A (**1**) isolated from the marine sponge *Axinella* sp. in 1999 [59] is the first compound to study. The second example is also a dimeric cyclic PIA from the marine sponge *Stylissa caribica*, tetrabromostyloguanidine (**2**) from 2007 [60], and the synthetic massadine derivative 3,7-*epi*-massadine chloride (**3**) is the last one of the PIA series from 2008 [61]. Finally, the terrestrial plant alkaloids tubocurarine (**4**) from *Chondrodendron tomentosum* [62] and vincristine (**5**) from *Catharanthus roseus* [63] are examples discussed here to illustrate the limitations of configurational analysis based on NOE/ROE data solely, and only the combined approach of using distance as well as RDC data allows to deduce their configurations unequivocally.

## 2. Results and Discussion

Compounds **1**–**3** are cyclic dimeric pyrrole-imidazole alkaloids (PIAs) with eight contiguous stereogenic centers each, resulting in 128 possible relative configurations (diastereomers), respectively. Axinellamine A (**1**) and 3,7-*epi*-massadine chloride (**3**) possess tetracyclic cores, whereas tetrabromostyloguanidine (**2**) features an even more complex hexacyclic core. For the PIAs **1**–**3** only ROE-derived interproton distances were used. The interproton distances were extracted from a ROESY spectrum with a mixing time of 100 ms (in case of **3**: 300 ms). For all compounds the interproton distances ±10% were used as distance restraints in the floating chirality restrained DG/DDD calculations (fc-rDG/DDD), additional details on the calculations on **1**–**3** are given in the Section 4 and the Appendix A. As NMR can anyhow determine relative configurations only, in all rDG simulations a single stereogenic center of **1**–**3** each was fixed by applying a chiral volume restraint in order to avoid enantiomeric structures. The number of the generated structures in the fc-rDG/DDD calculations was set to 1000 to allow for reasonable sampling of the configurational and conformational space. Additional simulations applying different chiral volume restraints and/or sampling lengths, as well as in-depth analyses of the rDG runs are described in the Appendix A. In the following, we report the application of the fc-rDG/DDD method to assign the relative configuration of all stereogenic centers for compounds **1**–**3** simultaneously, and based on ROE data alone.

### 2.1. Configurational Assignment with ROEs Only

#### 2.1.1. Axinellamine A (**1**)

For the configurational assignment of axinellamine A (**1**) 35 interproton distances from ROESY spectra were used (the complete list of ROEs of **1** is given in the Appendix A). As mentioned above, one stereogenic center of **1** was fixed and set as reference (C-14). In the traditional approach of pre-calculating structures, this would entail the necessity to evaluate a total of 128 diastereomers. Indeed, inspection of the output on the rDG protocol shows that all 128 configurations are actually generated by the “metrization” process in 4D space, but many of these molecular geometries severely violate the restraints imposed by the ROE data even in this higher dimension, and thus do not “survive” even the 4D refinement of simulated annealing. At the end of the 4D sampling phase, 40 alternative configurations were obtained (see Appendix A), out of which even only 37 did emerge finally from the 3D sampling, albeit many of these structures display severe ROE violations.

The over-all exceedingly high efficiency of configurational sampling by rDG, and the results for 1000 generated possible structural candidates of axinellamine A (**1**) are shown in Figure 4a (“best 700”) as a graphical representation of the total error (dimensionless) for each structure, ordered according to ascending total errors. The first wrong structure (wrong configuration of **1**) in respect to the eight stereogenic centers is No. #598 (red circle in Figure 4a). This structure differs from structures #1 to #597 by the configuration of C-1. The first “pseudo-configurational” change was already observed at structure No. #365 (orange circle in Figure 4a). This is the alternative assignment of the diastereotopic protons at the methylene group C-1′. Mathematically there is no difference between stereogenic and prochiral centers, which means that for axinellamine A (**1**) altogether ten centers needed to be assigned. Chemically only the stereogenic centers are of importance for the differentiation of the stereoisomers, but the prochiral centers are important to support the configurational assignment. In this example, only C-1′ is used, whereas the second prochiral center (C-1′’) does not contribute to the results since no ROE to both H’s of C-1′’ have been observed.

Most notably, the first wrong configuration of **1** (#598) appears rather late in this sequence of energy sorted structures sampled, visualizing the efficiency of sampling (the total number of structures with the correct configuration for axinellamine A (**1**) is even 760/1000). Additionally, the second best (first wrong) alternative configuration is separated from the best-fit global energy minimum structure by significant energy steps and a large pseudo energy difference of the penalty error function ($\Delta E_{total}=3.15$). Within the rDG approach, both of these characteristics are indicative for an unambiguous configurational assignment of **1** based on the experimental NMR data used, and the plot in Figure 4b shows, that all structures #1 to #597 indeed feature the same relative configuration of all stereogenic centers.

Figure 4a illustrates very well that the correct relative configuration of axinellamine A (**1**) appears in different conformations with respect to the orientation of the side chains. There are already six steps before a different configuration is observed, which originate from alternate local conformational changes that mainly include the orientation of the side chains (see Figure 4b). The inset plot in Figure 4a shows the first “energy” step in detail.

It must be stressed, that this rDG simulation is actually a single, fully automated sequence of calculations–and not 128 individual calculations on alternate diastereomers–by which the correct configuration of axinellamine A (**1**) is quickly and highly reliably identified. At no point of this simulation is a physical force-field involved, and the final assignment emerges based on experimental data only irrespective of the starting configuration.

#### 2.1.2. Tetrabromostyloguanidine (**2**)

For the configurational assignment of tetrabromostyloguanidine (**2**) 27 interproton distances from ROESY spectra were used (the complete list of ROEs of **2** is given in the Appendix A). Using the same methodology as in case of axinellamine A (**1**), the results for the 1000 generated possible structures of tetrabromostyloguanidine (**2**) are shown in Figure 5a (“best 400”) as a graphical representation of the total error (dimensionless) for each structure, ordered according to ascending total errors. As already discussed for **1**, one stereogenic center of **2** was again set as reference and fixed by the application of a chiral volume restraint (C-10). In order to verify and demonstrate that the results of the fc-rDG/DDD calculations do not depend on the choice of the stereogenic center that is fixed, these calculations were also repeated for all eight centers of **2** and are reported in the Appendix A (see Appendix A).

The first wrong structure with respect to the eight stereogenic centers is No. #378 (red circle in Figure 5). This structure differs from structures #1 to #377 by the configuration of C-20, which is actually the same position (different atom numbering, see Scheme 1) for the first change as observed in case of axinellamine A (**1**). The first “pseudo-configurational” change, i.e., an alternative diastereotopic assignment of methylene protons of the exocyclic methylene group C-19, was already observed at structure No. #99 (orange circle in Figure 5) with a very low energy difference of ΔE=0.14, indicating some ambiguity in the assignment of these CH_2_-protons. The second and more characteristic “jump” in energy is observed at structure No. #203. This jump in energy includes both either a new conformation of **2** and its side chains, or an alternative assignment of the diastereotopic protons at the endocyclic methylene group C-13, both changes have similar penalties in experimental versus calculated NMR parameters. At structure No. #277 the alternative assignment of the diastereotopic protons at C-13 is observed, and at structure No. #302 both methylene groups are inverted and both changes are manifested in rather small changes in pseudo energy only. The total number of structures with the correct configuration for tetrabromostyloguanidine (**2**) generated is 702/1000. Though all 128 relative configurations of **2** were initially generated by the rDG “metrization” step, only a few “survived” the 4D (36 configuration) and 3D (19 diastereomers) stages of sampling, and of the latter, all 18 wrong configurations appear after structure No. #377, and are ranked in their pseudo energy significantly higher (ΔE≥5.32) than the best-fit geometry of **2** with correct configuration (see discussion of **1**).

The diastereotopic assignment of the methylene protons can also be alternatively obtained by a *J* coupling approach (^3^*J*_HH_ and HMBC intensities). Using this information within the fc-rDG/DDD calculation, the results can still be improved (see Appendix A). For this calculation the two methylene groups are used with a fixed chiral volume, changing the number of floating centers from nine to seven (C-10 fixed). The first wrong structure for this calculation in respect to the eight stereogenic centers changes from No. #378 to No. #420, and many of the smaller steps in pseudo energy originating from alternate CH_2_-assignments vanish altogether.

In total, the relative configuration of all stereogenic centers in **2** is unequivocally determined by the ROE data used, although some ambiguities remain on the assignment of diastereotopic protons. However, both the unambiguity of the configurational assignment, as well as the ambiguity of the CH_2_-assignments is again established by a single rDG simulation, without any other assumptions or restraints used rather than experimental NMR data exclusively.

#### 2.1.3. 3,7-*epi*-Massadine chloride (**3**)

For the configurational assignment of 3,7-*epi*-massadine chloride (**3**) 36 interproton distances from ROESY spectra were used (the complete list of ROEs of **3** is given in the Appendix A). Results for the 1000 generated structures of 3,7-*epi*-massadine chloride (**3**) are shown in Figure 6a (“best 200”) as a graphical representation of the total error for each structure, ordered according to ascending total errors. As already discussed for **1** and **2** one stereogenic center of **3** was set as reference (C-13).

The first wrong structure in respect to the eight stereogenic centers is No. #56 (red circle in Figure 6). This structure differs from the preceding structures by a configurational change of C-3 and C-7, which represents the original massadine configuration. The first “pseudo-configurational” change was again observed earlier at structure No. #25 (orange circle in Figure 6a), which represents the alternative assignment of the diastereotopic protons at the methylene group C-1′’. The results of the calculations for **3** can be improved if a diastereotopic assignment of the methylene protons is carried out prior to the DG/DDD calculations (see discussion of compound **2**). In this case, the first wrong structure becomes No. #123 (see Appendix A). The diastereomeric differentiability in this case is not as pronounced as for compounds **1** and **2**. This becomes obvious just by looking at the occurrence of the first wrong structure (**1**: #597, **2**: #377, and **3**: #56), but it is still an unambiguous result. Another difference to the first two examples is the energy difference between the different configurations, which is much lower for **3**. This indicates that the extent and certainty with which the experimental data does differentiate between the different structures (diastereomers) is not as pronounced as it was observed for **1** and **2**. For 3,7-*epi*-massadine chloride (**3**), the NMR data set was less well defined because of the longer mixing time of the ROESY experiment.

### 2.2. Configurational Assignment with NOEs and RDCs

The terrestrial alkaloids tubocurarine (**4**) and vincristine (**5**) form the completion of the current investigation. In case of **4** only one stereogenic center relative to a second one needs to be assigned and is therefore a seemingly rather simple model for the described approach, but was chosen for demonstration purposes and its long-range separated stereogenic centers. Vincristine (**5**) is a more complex structure with nine stereogenic centers. In both examples, NOEs involving diastereotopic protons of methylene groups were used as unassigned and averaged restraints only, and in the case of **4**, the *o*/*o’*-protons of the *p*-disubstituted aryl ring were treated similarly (see Appendix A). As it will be demonstrated later in the manuscript, NOEs are not sufficient for the unambiguous assignment of the relative configuration of compounds **4** and **5**. Further data was necessary for a complete assignment. Due to the lack of experimental RDC data, we have decided to use synthetic RDC data sets for both compounds. Though one might argue that is a general weakness of the method, it is important to know for demonstration purposes that additional NMR parameter may help to solve the structural problem. Where applicable, RDCs involving CH_2_-groups were also treated as unassigned values, and only the sum of both individual ^1^*D*_CH_ methylene RDCs are used as restraining parameters. Though this reduces the quality of the data set that might be experimentally accessible, this method was chosen to reduce the amount of prior information to learn more about the limits of the DG based structure analysis described here. The full data sets of NOEs and RDCs used for **4** and **5** are listed in Appendix A. As this data explicitly does not allow to differentiate diastereotopic CH_2_-protons, we will not get into a debate on the subject of assignments on prochiral centers in this chapter.

#### 2.2.1. Tubocurarine (**4**)

The toxic alkaloid (“arrow poison”) tubocurarine (**4**) was chosen as a model compound with only two stereogenic centers (C-1 and C-24) being nine bonds apart from each other (either counting the orange or the blue pathway in the macrocycle; see Figure 7). Additionally, along either way only three out of the eight atoms in between have a proton attached, and thus **4** represents a prototype example where the relative configuration of the two remote stereogenic centers is expected to be indefinable on the basis of NOE data alone. The question is now: can RDCs contribute significantly to the assignment of the relative configuration of tubocurarine (**4**)?

For the configurational assignment of tubocurarine (**4**), a total of 17 NOE-derived interproton distances and 16 ^1^*D*_CH_ RDCs were used for up to three independent alignment media, respectively, the RDC data for **4** was taken from Ref. [22] (see also Appendix A).

Results for 1000 structures of tubocurarine (**4**) are shown in Figure 8a (“best 500”) as a graphical representation of the total error for each structure, ordered according to ascending total errors. Following the methodology outlined in the previous chapter for **1** to **3**, one stereogenic center of **4** was set as reference and fixed by a single rDG chiral volume restraint (C-1), and therefore, only one center needed to be assigned in the calculations. Using NOE data exclusively, the first wrong structure is No. #80 (black curve/circle in Figure 8a). The energy difference between the two structures of opposite configuration at C-24 is extremely low ($\Delta E_{total}=0.04$, see black symbols in Figure 8a). Accordingly, the total number of structures for tubocurarine (**4**) generated by rDG is almost equally distributed between both possibilities (the correct and the wrong configuration), and therefore a differentiation of the two alternative relative configurations of **4** by the NOE data set used here is impossible, as long as no further assumptions are made or additional experimental data is included.

The results can be significantly improved by adding RDC data to the restraints. A single alignment medium RDC data set with 16 individual ^1^*D*_CH_ RDCs added to the restraints of the rDG/DDD simulation leads to a clearly recognizable step in pseudo energy separating the first occurrence of a structure with wrong configuration (#334) from the energy minimum family of structures displaying the correct configuration of **4** (blue line and symbols in Figure 8a). This already pronounced diastereomeric differentiability is improved considerably when adding a second (Figure 8a, green line) or even third (Figure 8a, dark red line) RDC data set. Though these data sets require NMR measurements under different alignment conditions (alignment media) and are associated with quite some experimental effort, the resultant additional structural restraints add valuable information to the discrimination of diastereomers. With an increasing number M of alignment data sets used, the rDG/DDD simulations show a significantly increasing step in pseudo energy (*M* = 1: ΔE=0.95, *M* = 2: ΔE=1.73, and *M* = 3: ΔE=11.33, cf. Figure 8a) between both alternate configurational assignments of **4**, and the total number of correctly identified structures increases consistently. The second and third AM RDC data sets remove the last remaining doubts on the configuration of **4** that might prevail after single-AM analysis. The predictive power of AM data sets cannot be estimated in advance of a measurement, but needs to be evaluated thoroughly after the NMR data has been acquired. For the experimentalist, this is of high significance, as adding further data and re-running a rDG/DDD simulation is very straight-forward–it simply requires adding a new RDC table in an additional input file–and within a couple of minutes a clear-cut answer on the decidability of a given structural problem is provided by the DG method presented here.

The main plot of Figure 8a shows the combined usage of NOE and RDC data, whereas the inset graph reveals, that the discriminative power for alternate configurations based on RDCs alone is smaller than the combined usage of NOE and RDC data. Though the level of differentiation still increases with the number of alignment media data sets applied, the energy is smaller and less significant (*M* = 1: ΔE=0.41, *M* = 2: ΔE=0.41, and *M* = 3: ΔE=2.86, cf. Figure 8a, inset plot).

#### 2.2.2. Vincristine (**5**)

The alkaloid vincristine (**5**) from the pink-colored catharanthe (*Catharanthus roseus*) was chosen as very complex natural product. Vincristine (**5**) is an approved drug in cancer therapy. It has nine stereogenic centers, six of which are arranged consecutively in a six-membered ring and three are located in a remote ring fragment connected to the former segment by a single rotatable bond only, and therefore **5** is a challenging goal for a configurational analysis by NMR spectroscopy. For the configurational assignment of vincristine (**5**) altogether 23 NOEs and 24 RDCs in up to three AM, respectively, were used (all restraints were again used without assignments of diastereotopic methylene protons as described for **4**). The RDC data for **5** were taken from Ref. [22] for three independent alignment scenarios, respectively (see also Appendix A). It also must be noted that due to the absence of NOE and RDC associated with the substituents of C-42 (a quaternary carbon carrying a hydroxyl group and a COOMe ester moiety), the configuration of this stereogenic center is not assignable based on the data used here, and thus C-42 was excluded from any further analysis.

Results for 1000 structures of vincristine (**5**) are shown in Figure 9a (“best 100”) as a graphical representation of the total error for each structure, ordered according to ascending total errors. Again, a single stereogenic center of **5** was set as reference and fixed (C-9). Using NOE data only, the first wrong structure is already the structure No. #1 ranked best (black curve/circle in Figure 9a), which is a clear indication that NOEs alone are insufficient to accomplish the configurational analysis of vincristine (**5**). In analogy to the methodology employed in the case of **4**, the results can be improved by further adding RDC data to the restraints.

Successively adding multiple AM, RDC data sets slowly increases the certainty with which the correct configuration of **5** can be assigned: with *M* = 1 (blue curve), 2 (green), and 3 (dark red) AMs (see Figure 9a). However, the level of differentiability of the correct configuration from alternate wrong diastereomers is at first very low (*M* = 1: ΔE=0.04, and *M* = 2: ΔE=0.12), but raises constantly to ΔE=1.63 when using RDCs from three alignment media (*M* = 3). In the latter case, the first wrong structure is No. #87 (dark red triangle in Figure 9a), and this diastereomer is already separated from the best-fit (pseudo energy minimum) correct diastereomer of **5** by a now significant step in the error function, which is due to a configurational change of the quaternary carbon C-17 (at structure #87). Another step (not shown) follows at structure #245 (ΔE=6.04) due to a misassignment of C-41. In conclusion, the most problematic stereogenic centers to be determined for vincristine (**5**) are–as discussed above–C-42, C-17, and C-41 (in this order), whereas the reliability with which any of the other six stereogenic centers is differentiated from alternate configurations is high when NOE and RDC data is used in combination (see Figure 10 for a traffic-light type encoded pictorial description of these assignment probabilities). The problems associated with C-17 arise from limited RDC data available for the rotating ethyl side chain, and for C-41, only a single CH RDC and two NOEs indicate some preference for the correct configuration over a wrong assignment. However, Figure 9a and the inset plot therein clearly indicate the importance of using combinations of both NOE and RDC data sets, as neither NOEs nor RDCs alone provide conclusive evidence for the correct configuration of **5**, and in particular calculations relying on RDCs only gave much less conclusive results as compared to the combined approach.

## 3. Conclusions

In this study, we have shown with the aid of five examples of natural products, that the ROEs or NOEs/RDCs driven floating chirality distance geometry (fc-rDG/DDD) approach represents a valuable method to assign the configuration (and conformation) of complex molecules in just one single calculation. Given the known constitution of a compound, the method produces all configurations that are in accordance with the experimental NMR data, without the necessity to carry out separate configurational and conformational analyses on $2^{n-1}$ diastereomers for n stereogenic centers. In the case of the marine natural products **1**–**3**, the relative configuration of eight stereogenic centers is unequivocally derived in just one instead of individual 128 simulations. In addition, it was demonstrated for the terrestrial alkaloids **4** and **5**, that the DG method also clearly reveals remaining ambiguities if NOE data alone is insufficient for configurational assignments–as e.g., for the long-range separated stereogenic centers of **4**–and indicates to the NMR spectroscopist that additional data such as RDCs has to be acquired. Successively adding RDC data obtained for different alignment media as additional restraints to the DG calculations is straight forward, and can be easily repeated until the level of confidence of the assignment is raised beyond any reasonable doubt.

The method discussed here neither requires individual treatment of alternate diastereomers under consideration, nor does it rely on force-field based MM/MD or DFT derived pre-calculated structures. In particular the use of force-field parameters–which may not even be available for uncommon structural fragments of natural products–in the traditional MD approach introduces an implied bias towards low-energy structures (in a thermodynamic sense) that might be misleading as in the case of tetrabromostyloguanidine (**2**). Here, the correct configuration of two *trans*-anellated five-membered rings is about 24 kJ mol^−1^ less favorable than the more stable wrong configuration with *cis*-fused rings (which represents the original palau’amine configuration from 1993 [64,65,66,67,68]; see Figure 11), and any MD approach would have to overcome this pronounced bias in order to identify the correct configuration of **2**.

Most importantly, the methodology outlined here does not depend on pre-calculated structures that are traditionally evaluated against experimental data, but DG represents the opposite approach, which produces structures that evolve from experimental restraints exclusively. The “FF- and DFT-free” types of simulations are unbiased, reliable, and fast (completed within minutes), and the NMR data itself governs the mode with which these structures emerge.

The full methodology outlined here for the interpretation of NOEs/ROEs and RDCs has been implemented in our ConArch^+^ (Configurational Architect) program, which also produces convenient pseudo energy and configuration sorted lists that were used for all plots presented here. The software can be obtained along with the source code (free of charge for academic institutions) from our web site (https://www.chemie.tu-darmstadt.de/reggelin).

## 4. Methods

### 4.1. NMR Data

The ROEs for compounds **1**–**3** were taken from refs [60,69]. For compounds **1** and **2** three ROESY spectra with different mixing times (100, 150 and 200 ms) were measured [60,69]. In the case of **3**, only one ROESY spectrum with a mixing time of 300 ms was recorded [69]. The spectra were integrated with TOPSPIN and SPARKY.

For the compounds **4** and **5**, sets of NOEs were predicted using the corresponding X-ray structures and all proton-proton contacts ≤3.5 Å. All NOEs involving CH_3_– or CH_2_– groups were treated as averaged values between unassigned (diastereotopic) protons only, in order to reduce the amount of prior information, thus simulating situations where no diastereotopic assignment was possible. The *ortho*- and *ortho*’-protons of the central benzene ring (C-17 to C-22) were treated equivalently (see Appendix A). Simulated RDC data for **4** and **5** was taken from Ref. [22] for three independent alignment scenarios, respectively.

### 4.2. DG/DDD

The ROE-/NOE-interproton distances as well as the RDCs served as input for the distance geometry (DG) and distance bounds driven dynamics (DDD) calculations. The DG pseudo force-field employed for all simulations presented in this study takes the form defined by Equation (1):(1)$$E_{total}=E_{dist}+E_{chir}+E_{NOE}+E_{RDC}+E_{DBL},$$
where the dimensionless total pseudo energy $E_{total}$ is a sum of distance (holonomic bond lengths) errors ($E_{dist}$), chiral volume violations ($E_{chir}$), NOE ($E_{NOE}$), and RDC ($E_{RDC}$) deviations of experimental data from values back-calculated from structures and a special term denoting the deviations of double bonds from planarity ($E_{DBL}$). There are no additional or customary atom-type dependent force-field parameters of physical force-fields used. All pseudo energy terms take the form of sums of squared violations ${\left(\Delta X\right)}^{2}$ as defined by Equation (2):(2)$$E_{X}=\frac{1}{2}K_{X}\sum {\left(\Delta X\right)}^{2},$$
with $\Delta X=X^{exp}-X^{calc}$, and empirically chosen force-constants $K_{X}$ to appropriately account for the size and allowed ranges of each type of parameter violations ΔX.

$E_{dist}$ and $E_{chir}$ (Equation (1)) represent the violations originating from differences in holonomic distances $\Delta r_{i,j}$ (i.e., bond lengths) and $\Delta V_{i}$ (chiral volumes). The latter are defined by the scalar triple product $V_{chir}=\vec{a}\cdot \left(\vec{b}\times \vec{c}\right)$ of three vectors spanning planar sp^2^-type ($V_{chir}=0$) or tetrahedral sp^3^-type atomic centers (i.e., stereogenic centers with $V_{chir}\neq 0$), thus encoding for the configuration of the latter through opposite signs ($\left|V_{chir}^{\left(S\right)}\right|=\left|V_{chir}^{\left(R\right)}\right|$), respectively (see Figure 12a). For reference, holonomic distance bounds (for all atom-atom pairs for which upper and lower bounds of inter-atom distances can be established based on the molecular constitution) and chiral volumes are obtained from an initial guess (input) structure of arbitrary configuration and conformation. As these values depend solely on the constitution (which must be known), the DG approach is completely independent from the structure initially assumed [22]. Here, chiral volume restraints were used only on a single stereogenic center simply to avoid enantiomeric structures, as well as on CH_3_-groups (to keep them tetrahedral) and all sp^2^-centers (to keep them planar, $V_{chir}=0$). Thus, through the deliberate absence of chiral volume restraints, all stereogenic centers (except one), and all CH_2_-groups with diastereotopic protons were allowed to “*float*” and thus their configurations and/or assignments evolve on the basis of experimental NMR data (NOEs and/or RDCs) only.

In Equation (1), $E_{NOE}$ denotes the deviations of back-calculated (and $<r^{-6}>$ averaged where applicable) NOE distances from experimental upper and lower distance bounds with Δr defined as follows:(3)$$\Delta r=\{\begin{array}{ccc} \;r_{i,j}-r_{lower} & \mathrm{for} & r_{i,j}<r_{lower}\\ 0 & \mathrm{for} & r_{lower}\leq r_{i,j}\leq r_{upper}\\ \;r_{i,j}-r_{upper} & \mathrm{for} & r_{i,j}>r_{upper} \end{array}.$$

In this study, all NOE bounds $r_{lower}$ and $r_{upper}$ were derived from the corresponding NMR volume integrals and used as $r_{mean}\pm 10\%$, with force-constants $K_{NOE}=10.0$ Å^−2^ unless stated otherwise.

The mathematics of RDC calculations used here has been taken from Glaser et al. [53], and the formalism on how to include RDC data in 4D and 3D DG simulations (see below) has been described in full detail in Ref. [22]. The harmonic RDC “pseudo energy” $E_{RDC}$ is based on violations $\Delta D=D^{exp}-D^{calc}$ between experimental and back-calculated values, and RDC data sets derived from multiple alignment media can be used simultaneously as an increasing number of experimental NMR restraints in our *ConArch*^+^/*DG* approach simply by expanding the corresponding sum in $E_{RDC}$. Empirically, it proved best to scale the force-constant $K_{RDC}$ used with the number of RDCs, or equivalently, with the number M of alignment media used, and thus we employ $K_{RDC}=1.5/M$ Hz^−2^ in this study.

In addition to the application of chiral volume restraints on sp^2^-type atomic centers ($V_{chir}=0$), the term $E_{DBL}$ in Equation (1) is used to reasonably restrain double bonds and aryl rings to planarity (restraining $V_{chir}=0$ on neighboring atomic centers alone is not sufficient). Here, ΔX (used cf. Equation (2)) is defined as $\Delta X=1-\cos^{2}\phi$, where $\phi_{i,j,k.l}$ are the corresponding torsion angles i-j-k-l with sp^2^-sp^2^-type central bonds, and ΔX vanishes for ϕ=0° and ϕ=180° only (*cis*- and *trans*-configurations). The rather high force-constant $K_{DBL}=100$ used here efficiently removes local energy minima which originate from slight bending of C=C-double bonds and aromatic rings, revealing more distinctive energy steps separating alternate conformational families. In general, the final best-fit energy-minimum structures have very low distortional energy terms of $E_{DBL}\leq 5\times 10^{-3}$, and the efficiency with which different configurations and conformations are sampled on the basis of experimental NMR data is largely unhampered by these types of restraints.

The initial input structure is used by DG only for setting up the holonomic bounds and distance matrices (±1% bond lengths), and subsequent configurational and conformational sampling is carried out by our ConArch^+^/DG approach in an automated sequence of steps. First, molecular structures are generated in four-dimensional (4D) space (“metrization” step, i.e., embedding based on holonomic distance bounds), followed by a 4D “floating-chirality” restrained DG (fc-rDG) and distance bounds driven dynamics (DDD) simulation (simulated annealing). After reduction of dimensionality, the simulated annealing is repeated in 3D space, and each simulation in 4D and 3D is concluded by a gradient-descent type optimization of structures against all restraints, minimizing the total pseudo energy $E_{total}$. In all dynamics and optimization calculations, the partial derivatives $\partial E_{total}/\partial r_{\alpha }$ of all energy terms with respect 4D and 3D Cartesian atomic coordinates (α∈x,y,z(,w) for all atoms) are interpreted and used as forces governing the evolution of the system. All derivatives are calculated analytically by ConArch^+^/DG. During each step of the rDG/DDD runs using RDCs, full updates of the Saupe or alignment tensors are computed based on a singular value decomposition (SVD) algorithm.

Sampling molecular structures first in 4D very efficiently generates diastereomeric geometries as inversion barriers can be overcome easily [70,71]. Configurational inversion in 3D is reduced to a simple rotation in 4D (see Figure 12b), and consequently during simulated annealing in 4D space with chiral restraints removed on all but selected chiral centers (fc-DDD), the transition barriers between diastereomers are significantly lowered or removed altogether.

For an increased sampling efficiency, it is crucial to transport as much 4D information as possible into 3D, in order to produce chemically relevant structure models. Projection from higher to lower dimensionality optimally preserves atom-atom distances when carried out along the eigenvector associated with the largest eigenvalue of the inertia tensor I defined by Equation (4):(4)$$I=\sum_{k}\left(\left({\vec{r}}_{k}\cdot {\vec{r}}_{k}\right)E-{\vec{r}}_{k}\;⨂\;{\vec{r}}_{k}\right),$$
where the sum runs over all atomic positional vectors (${\vec{r}}_{k}$ for k particles) centered about the origin ($\sum {\vec{r}}_{k}=0$) and weighted with unity mass (see Figure 12c) [72]. During 4D simulated annealing, we apply a temperature dependent scaling factor $f=\exp\left(-{\left(T/\tau_{T}\right)}^{2}\right)$ with an empirical temperature coupling factor $\tau_{T}=150$ K to forces acting along this eigenvector, which gradually restrain the 4D molecular models into a 3D subspace thereof (see Figure 12d). At high temperatures (T>300 K), all structures evolve freely, but are restrained increasingly and smoothly to 3D sub-space during the cooling phase (T<300 K) of the simulated annealing. Finally, all models are projected into pure 3D space and are subjected to an additional simulated annealing therein.

In this study, for each compound **1**–**5** a total of 1000 structures (configurations and conformations) were generated initially in 4D space. All consecutive simulated annealing simulations in 4D and 3D used 5000 steps of equilibration (T=300 K) and 5000 steps of cooling (T→0 K) each (2 fs time steps). The final structures were collected, sorted by their pseudo energy, and a final selection of the ranked structures of lowest energy were used for the plots presented here. For the global energy minimum best-fit structures, errors in calculated RDCs are estimated from Monte-Carlo bootstrapping analysis including tensor updates [22,23]. Total single processor CPU (Intel(R) Core(TM) i7-4790 CPU @ 3.60GHz) wall time used was about 7–8 min. for each compound **1**–**3**, and up to approx. 50 min for **4** and **5** when three alignment media RDC data sets are used. However, the entire process can be parallelized very efficiently on an arbitrary number of shared memory CPU cores, reducing the total wall time accordingly to a few minutes only.

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
