# Peer review of "The Advanced Floating Chirality Distance Geometry Approach―How Anisotropic NMR Parameters Can Support the Determination of the Relative Configuration of Natural Products"

_marinedrugs, 2020, doi:10.3390/md18060330_

Round 1
Reviewer 1 Report
The manuscript by M. Kock et al. reports about an interesting methodology to assign the relative configuration and conformation of complex molecules in just one single calculation, showing a big potential of application. For this aim five natural products, three from marine and two from terrestrial source, have been selected. The article is well written and interesting to read. For these reasons I believe that the manuscript could give a substantial contribution to the field of configurational analyses of marine natural products. Therefore, the publication of this article on Marine drugs can be considered after the following minor revision:
Page 1, lines 24: ‘Compounds 1 to 3’, please correct in ‘compounds 1-3’.
Several words in the manuscript are underlined, please correct.
In the first part of introduction, add a more recent reference regarding the recent methods used to solve stereochemistry issues of natural products (Menna et al. Nat. Prod. Rep., 2019, 36, 476).
Page 9, line 335: Please correct the paragraph title
Page 9, line 339: ‘As already discussed for1’. Please, add a space.
Page 12, line 452: ‘from Ref. [20]’. Why in red??
In the main text not all tables and figures reported in SI are mentioned.
Check the formatting of the reference section
Author Response
1-1) Page 1, lines 24: ‘Compounds 1 to 3’, please correct in ‘compounds 1-3’.
Since this is the start of the sentence the first letter should be capitalized. If the reviewer wants that “to” should be changed to “-“ we should leave it to journal and the editor to change it to the journal’s format.
1-2) Several words in the manuscript are underlined, please correct.
We wanted to emphasize certain words. First we had the idea to capitalize these or print them in Italics. Version 1 seemed to be too aggressive whereas version 2 was not really different from the rest of the text. Therefore, we have decided to underline certain words throughout the manuscript.
1-3) In the first part of introduction, add a more recent reference regarding the recent methods used to solve stereochemistry issues of natural products (Menna et al. Nat. Prod. Rep., 2019, 36, 476).
The reference was added to the manuscript as requested by the reviewer.
1-4) Page 9, line 335: Please correct the paragraph title
The paragraph title was corrected.
1-5) Page 9, line 339: ‘As already discussed for1’. Please, add a space.
A space was added.
1-6) Page 12, line 452: ‘from Ref. [20]’. Why in red??
There was no reason for that. It was changed to black.
1-7) In the main text not all tables and figures reported in SI are mentioned.
We have checked the whole text again in this respect and added references at the appropriate positions in the text.
1-8) Check the formatting of the reference section
There were some problems with the format of the references which we have changed as requested by the reviewer.
Reviewer 2 Report
The Authors have completed a very good a piece of work. The perfect combination of ROEs or NOEs/RDCs driven floating chirality distance geometry approach represents notably a method to assign the configuration of molecules in just one single calculation. I think, the ConArch+ program will be used by many institutions.
Author Response
No specific comments.
Reviewer 3 Report
To enhance the breadth of the introduction.
Lines 142-146- Residual dipolar couplings can arise even in solvents with isotropic environment when the molecule has significant intrinsic magnetic anisotropy as has been amply demonstrated for molecules (even proteins) containing paramagnetic metal ions.
Concerning the results
Lines 173-176 Large changes in chemical shift with the use of anisotropic media may indicate structural changes which may raise doubts on the relevance of the structure obtained with this method. The authors may wish to discuss thisaspect.
Line 233- which are these 35 interproton distances? This information should be made available in the supplementary materials to allow the confirmation that there is adequate coverage of the molecule. This is important information as become evident in lines 254-255.
Finally,
Lines 551: this in my opinion is an important weakness of the manuscript. The authors did not use experimental data as source of information for RDCs but used instead simulated data. A justification for this choice and a discussion of the consequences should to be made.
Author Response
3-1) Lines 142-146- Residual dipolar couplings can arise even in solvents with isotropic environment when the molecule has significant intrinsic magnetic anisotropy as has been amply demonstrated for molecules (even proteins) containing paramagnetic metal ions.
This was changed as requested by the reviewer.
We changed the passage to:
“Standard NMR investigations are carried out in isotropic solutions, where usually the dipolar couplings are averaged out by isotropic tumbling of the molecules. If this is not the case, either by the presence of paramagnetic metal ions[1] or anisotropic susceptibility of diamagnetic macromolecules[2] or, more general, the presence of an anisotropic medium, the molecules will be partially oriented with respect to the external magnetic field, and residual dipolar couplings (RDCs) can be measured (detailed reviews can be found at XXX).”
… and added 2 more references, clarifying this point.
- Tolman, J. R.; Flanagan, J. M.; Kennedy, M. A.; Prestegard, J. H., Nuclear magnetic dipole interactions in field-oriented proteins: information for structure determination in solution. Proceedings of the National Academy of Sciences 1995, 92, (20), 9279-9283.
- Tjandra, N.; Grzesiek, S.; Bax, A., Magnetic Field Dependence of Nitrogen-Proton J Splittings in 15N-Enriched Human Ubiquitin Resulting from Relaxation Interference and Residual Dipolar Coupling. J. Am. Chem. Soc. 1996, 118, (26), 6264-6272.
Concerning the results
3-2) Lines 173-176 Large changes in chemical shift with the use of anisotropic media may indicate structural changes which may raise doubts on the relevance of the structure obtained with this method. The authors may wish to discuss this aspect.
Yes, indeed such shifts could indicate conformational changes, but in the approach used here this does not affect the determination of the relative configuration. The distance geometry based approach as opposed to the usual procedure of comparing RDCs back-calculated from DFT-structures with the experimental ones does not rely on these structures. On the contrary: The experimental restraints let the structures evolve irrespective of the starting structure. Therefore, any conformational change that may occur as consequence of the interaction of the analyte with the medium will be found together with the configuration. It is a simultaneous determination of both structural aspects (configuration and conformation) without bias related to the input structure (which – apart from the constitutional information - is lost in the early stage of the calculation anyway).
This was not added to the manuscript! But this could be changed if the editor wants to add the comments to main text.
3-3) Line 233- which are these 35 interproton distances? This information should be made available in the supplementary materials to allow the confirmation that there is adequate coverage of the molecule. This is important information as become evident in lines 254-255.
The complete list of ROEs for 1 is given in the SI. Therefore, the required information is already included. We have added a reference to the SI at the positions in text as it was requested by the editor.
3-4) Lines 551: this in my opinion is an important weakness of the manuscript. The authors did not use experimental data as source of information for RDCs but used instead simulated data. A justification for this choice and a discussion of the consequences should to be made.
The problem is that are only a few experimental RDC data sets of natural products. Since we did not want to repeat the work on IPC or strychnine (for which we have experimental RDC data sets) we have decided to use simulated data (for compounds 4 and 5).
The first point was also argued in a very recent review on the stereochemistry of natural products (Menna et al. Nat. Prod. Rep. 2019, 36, 476).
“In spite of their great potential, the use of RDCs and RCSAs is not yet a routine method for stereochemical assignment.”
“Finally, RDCs and RCSAs have mostly been used to study conformationally rigid compounds.19,23 There are still limits to the their applicability to flexible molecules, especially when the stereo-centers to be correlated are linked through a flexible chain.”
This was not added to the manuscript! But this could be changed if the editor wants to add the comments to main text.
Round 2
Reviewer 3 Report
No further comments to the authors
Author Response
I did not see any comment in the review report of reviewer 3 (round 2).